# Nutrients, Phytic Acid and Bioactive Compounds in Marketable Pulses

**DOI:** 10.3390/plants12010170

**Published:** 2022-12-30

**Authors:** Lovro Sinkovič, Barbara Pipan, Filip Šibul, Ivana Nemeš, Aleksandra Tepić Horecki, Vladimir Meglič

**Affiliations:** 1Crop Science Department, Agricultural Institute of Slovenia, Hacquetocva Ulica 17, SI-1000 Ljubljana, Slovenia; 2Department for Chemistry, Biochemistry and Environmental Protection, Faculty of Sciences, University of Novi Sad, Trg Dositeja Obradovića 3, RS-21000 Novi Sad, Serbia; 3Faculty of Technology, University of Novi Sad, Bulevar Cara Lazara 1, RS-21000 Novi Sad, Serbia

**Keywords:** seeds, pulses, protein, multi-mineral profile, phytic acid, phenolics

## Abstract

Pulses are edible seeds of plants belonging to the legume family, which are of great importance for human and animal nutrition. In this study, several nutrients, antinutrients and bioactive compounds were quantified in the seeds of ten pulses, i.e., common and runner beans, field peas, lupins (white, blue and yellow), faba beans, lentils (brown and red) and chickpeas. Homogenised, air-dried seed samples were analysed for various parameters: protein (18.0–43.1%), fat (0.6–18.5%) and phytic acid content (507–2566 mg/100 g dry weight (DW)), phenolic profile (27 phenolic compounds in total) and multi-mineral composition. The analysed phenolic compounds mainly belong to phenolic acids (hydroxybenzoic acids and hydroxycinnamic acids) and/or flavonoids (flavones, flavonols and flavanols). Total phenolic content (TPC) ranged from 719 μg/g DW in chickpeas to 5012 μg/g DW in common beans. A total of ten elements belonging to macro- (Mg, P, S, K and Ca) and micro-minerals (Cr, Mn, Fe, Zn and Mo) were determined. Using cluster analysis, pulses were divided into three groups according to the parameters studied: 1. common and runner bean; 2. field pea, white and blue lupin, faba bean, red and brown lentil and chickpea; and 3. yellow lupin. The most varying phytochemicals in terms of their content in the analysed pulses were phytic acid, quinic acid, catechin and TPC. A perfect positive significant Pearson correlation (1.00) was observed for six pairs of variables within the group of phenolic compounds.

## 1. Introduction

Pulses are annual leguminous crops (Leguminosae family) yielding grains or seeds used for food, feed and sowing purposes. The denomination pulses are limited to crops harvested for dry grain only and exclude crops harvested green for food (e.g., green beans, green peas), forage, grazing or as green manure [1]. Annual legumes are important players in sustainable agriculture. Their ability to form a symbiosis with rhizobia, resulting in nitrogen fixation, contributes to increasing levels of nitrogen in the soil, which limits the use of nitrogen fertilizers. In addition, crop rotation and intercropping with legumes—which are also of particular interest in organic agriculture—improve soil fertility and yield and help control weeds and pests [2].

Given the world population growth, food supply is and will be one of the most significant challenges of the future. Therefore, switching to a plant-based diet and reducing meat consumption are considered good strategies to alleviate this problem. Pulses, as nutrient-rich food with a low environmental footprint (lower impact on climate, land use and nitrogen fertilization), could be one of the key drivers for this dietary shift [3].

As a food source, pulses are considered rich in protein, dietary fibre and complex carbohydrates [4]. The high protein content (17–30% per dry weight) makes pulses an excellent source of protein, in addition to their high content of essential lysine amino acid, which is low in cereals. Although carbohydrates account for 50–65% of their weight, they are considered low glycemic index foods due to their long digestion time. In addition, although pulses have a low total fat content, they are a good source of monounsaturated and polyunsaturated fats and sterols. In addition to macronutrients, pulses are also rich in micronutrients such as minerals (potassium, zinc and iron) and vitamins (B1, B2, B3, B6, B9, A and E). Finally, phenolic compounds—secondary plant metabolites—are also present in pulses [5]. Phenolic acids, flavonoids and condensed tannins are distributed in the seed coat (particularly flavonoids) and cotyledon (mainly phenolic acids) of legumes [6]. The nutritional factors in pulses are widely considered indispensable in human consumption, but these foods also provide several non-nutritive bioactive compounds that act as natural antioxidants. Phenolic compounds are one of the largest groups of non-essential dietary constituents whose bioactivity is attributed to their ability to chelate metals, inhibit lipid peroxidation and scavenge free radicals [7]. When it comes to phenolics in plant foods, flavonoids are usually the most abundant class and account for about two-thirds of the phenolic compounds in the diet. Phenolic acids are present in plant foods mostly in bound form: either as simple glycosides, as insoluble structural components of the cell wall in combination with xylans, pectin and lignin or as conjugated esters [8]. The most common hydroxycinnamic acids are caffeic acid, *p*-coumaric acid and ferulic acid, which are often found in plant foods as simple esters with quinic acid or glucose [9]. Quinic acid is not a phenol because it has no phenolic chemical structure, but it can play an important role in the formation of phenolic compounds [10]. The seed coat of pulses serves as a protective barrier for the cotyledon and contains a high concentration of phenolic compounds [6]. The use of polyphenol-rich pulses as functional foods could have a positive impact on human health, as these compounds are known to be potent antioxidants that protect against oxidative stress and degenerative diseases [11]. Pulses are thought to have health-promoting effects on blood lipid profile, blood glucose control, inflammatory status and oxidative stress, which are major risk factors for cardiovascular disease [5]. Research suggests that introducing pulses into the diet may prevent hyperlipidemia and hypertension [12].

Pulses are also a source of various bioactive compounds such as enzyme inhibitors, lectins, phytates and oligosaccharides, which may have beneficial or detrimental effects when ingested by humans or animals. Several compounds such as trypsin and chymotrypsin inhibitors, phytic acid, tannins and oligosaccharides (raffinose, stachyose and verbascose) are considered dietary inhibitors or antinutritional factors because they limit protein and carbohydrate utilization [4,13]. Antinutritional effects of the non-protein compound phytic acid have been associated with impaired absorption of elements and deterioration of lipid and protein absorption [14]. Phytic acid is a negatively charged molecule that binds zinc, iron, magnesium, phosphorus and calcium [14,15]. Raffinose, stachyose and verbascose are predominant α-galactosides present in pulses. These oligosaccharides are not absorbed or hydrolysed by the human intestinal microbiota due to the absence of an endogenous α-galactosidase enzyme. Therefore, the gasses fermented during the degradation of these carbohydrates by colonic bacteria may cause bloating and abdominal discomfort when pulses are consumed [11]. Nowadays, anti-nutritional factors such as saponins, tannins, phytic acid, lectins, protease inhibitors, amylase inhibitors, etc., are only a minor problem because there are strategies to reduce them. These include processing procedures such as milling, soaking, germination, autoclave and microwave treatment, and fermentation [15,16].

The intra- and inter-species diversity of pulses is consistently associated with differences in their composition and potential content of health-promoting compounds. Due to the growing importance of pulses, we investigated the phytochemical profile of common and runner beans, field peas, white, blue and yellow lupins, faba beans, red and brown lentils, and chickpea seeds. These marketable pulse seed samples were analysed for nutrients (crude protein, crude fat and multi-mineral composition), antinutrients (phytic acid content), and bioactive compounds (individual and total phenolics). The aim of our study was to compare the results obtained between and among the selected species. Another objective was to evaluate correlations between the analysed compounds and to perform a cluster analysis.

## 2. Results and Discussion

### 2.1. Protein, Fat and Phytic Acid

The pulses examined in this study yield one to eight grains or seeds of varying size, shape and colour in a pod. Common bean, field pea and faba bean developed a greater number of seeds (average of four to five) in the pods than runner bean and lupins (average of two to three). Chickpea and lentil generally develop smaller pods with one to two seeds. The seed colours of common and runner bean were brown, white, black, or a mixture of colours—a combination of red and beige. Their nutritional and phytochemical composition is very diverse, so they are the subject of numerous research projects. Pulses are considered a low glycemic index food and an element of the Mediterranean diet [13]. The nutritional properties and phytic acid content in the seeds of the studied pulses are shown in Table 1. Moisture content was highest in field peas (15.6%), faba beans (14.4% and 13.5%) and white lupin (13.5%). Crude protein content ranged from 18.0% to 43.1% and crude fat content from 0.6% to 8.5%. Crude protein content was highest in yellow lupin (43.1%) and lowest in runner bean (18.0%). Pulses with protein content above 30% were white, blue and yellow lupins, and with protein content below 20% were chickpeas, field peas and runner beans. In general, lupins had the highest protein content (32–43%) among pulses, while protein content varied slightly among different accessions/varieties of common beans (20–24%). Protein content of pulses generally ranges from 15–30% [17]. Similarly, Boeck et al. [18] reported pulses with protein contents of 20–30%. Hall et al. [17] described crude protein contents of 36% for faba beans, 19–27% for chickpeas, 23–31% for lentils, 14–31% for peas, 32–44% for lupins and 17–28% for beans. From a nutritional perspective, pulses are among the richest dietary sources of protein and amino acids for human nutrition [14]. Pulses proteins contain more essential amino acids, especially lysine, compared to animal proteins [19].

Crude fat content was highest in white lupin (8.5%), followed by chickpea (5.4%) and blue lupin (5.4%), and yellow lupin (4.8%). The common bean sample SRGB204 (0.6%) had the lowest crude fat content. The crude fat content of common beans, field pea, faba beans, red lentils and brown lentils was 1.6% or lower. In general, lupins (white, blue and yellow) and chickpea contained the highest fat content (5–9%) among the pulses studied. From a nutritional point of view, the fatty acid composition of pulses consists of about 50% polyunsaturated fatty acids (PUFA), with the essential linoleic acid and linolenic acid present in large amounts, 30% monounsaturated fatty acids (MUFA) with the main part oleic acid, and 20% saturated fatty acids (SFA) with the main part palmitic acid [14]. The lipid content of pulses is generally less than 3%. However, the lipid content of chickpeas and lupins can be as high as 5% and 13%, respectively [17]. Caprioli et al. [20] conducted a comprehensive assessment of lipid extraction in various pulses and found that some fatty acid contents were higher than previous reports, likely due to the extraction protocol. Hall et al. [17] reported crude fat contents of 2–7% for chickpea, 1–3% for lentil, 1–4% for pea, 5–15% for lupin and 1–4% for bean. Higher crude protein and crude fat contents in lupins compared to other pulses have been reported previously [17,18]. The obtained results on crude protein and crude fat content are in agreement with literature data on various pulses or grain legume species [14,21,22].

Phytic acid, known as an antinutrient of non-protein origin, is capable of complexing micro- and macroelements, even reducing the bio-functions of minerals and proteins or complexing enzyme ion cofactors [14,23,24]. However, it has been found that phytic acid in lower concentrations may have some beneficial nutritional effects, such as lowering blood glucose levels, combating dental caries, preventing colon cancer, antioxidant activity, etc. [25]. The phytic acid content of pulse species studied varied widely, ranging from 507 mg/100 g DW to 2566 mg/100 g DW. The highest levels were found in common bean samples SRGB204 (2566 mg/100 g DW) and Golden gate (2550 mg/100 g DW) and the lowest in faba bean sample Merkur (507 mg/100 g DW) and brown lentil (612 mg/100 g DW). Pulses with phytic acid content higher than 1700 mg/100 g DW were the common bean samples except for Etna, field pea, and white and yellow lupin. Lower phytic acid levels (<1200 mg/100 g DW) were found in the runner bean, one faba bean sample, red and brown lentils, and chickpea. The phytic acid content in different pulses ranged from 0.27% to 2.90%, which is consistent with the data obtained here [18,26]. The ideal phytate content for healthy intake may be 25 mg or less per 100 g in the consumed diet to minimise micronutrient losses. Although the level of phytic acid was much higher in the pulses studied, it can be reduced by soaking and cooking [19].

### 2.2. Phenolic Profile

The identified and quantified compounds in the seeds (including seed coats) of common bean, runner bean and field pea are listed in Table 2, and those of white, blue and yellow lupins, faba bean, lentil and chickpea are listed in Table 3. Of the compounds analysed, only quinic acid, amentoflavone and ferulic acid were present in higher concentration (above the LoQ) in all pulse samples tested.

Common bean samples showed wide variation in the presence and content of quantified phenolic compounds (Table 2). Quercetin 3-O-glucoside was the most abundant compound in the SRGB Škobrne (1958 μg/g DW) and SRGB196 (1804 μg/g DW) common bean samples, whereas it was not present at all in the Golden gate sample. Catechin was a relatively abundant compound (>421 μg/g DW) in four common bean samples and absent in two samples. Baicalein, epicatechin, isorhamnetin and luteolin 7-*O*-glucoside were present in only three samples of common bean, with the latter present at lower concentrations (5–6 μg/g DW). Quercetin was present in higher amounts in two samples of common bean SRGB Škobrne (494 μg/g DW) and SRGB196 (379 μg/g DW) compared to other samples. Kaempferol 3-*O*-glucoside varied significantly among the four common bean samples (9–436 μg/g DW). Rutin, with a complex phenolic structure, was present at low concentrations in five common bean samples (6–46 μg/g DW) and *p*-coumaric acid was present in all common bean samples (1–11 μg/g DW). The compounds kaempferol 3-*O*-glucoside (77 μg/g DW), gallic acid (32 μg/g DW) and caffeic acid (10 μg/g DW) were each present in only one common bean sample. *p*-Hydroxybenzoic acid was present at low concentrations (3 μg/g DW) in two common bean samples. High variability of catechin (0–614 mg/kg DW), epicatechin (0–279 mg/kg DW) and kaempferol 3-*O*-glucoside (0–1486 mg/kg DW) content among different common bean (*Phaseolus vulgaris*) types has been previously reported [27,28]. Phenolic profiles showed greater variability in common bean and faba bean cultivars compared to other pulses such as chickpeas, lentils and peas in previous studies [29].

The most abundant compounds in runner bean and field pea were quinic acid and catechin (Table 2). Quinic acid at 992 μg/g DW and 545 μg/g DW, and catechin at 216 μg/g DW and 202 μg/g DW, respectively. Phenolic acids such as gallic acid, *p*-hydroxybenzoic acid and *p*-coumaric acid, and flavonoids such as quercetin 3-*O*-glucoside and rutin were also found in runner bean. Similarly, phenolic acids such as caffeic acid, *p*-hydroxybenzoic acid and *p*-coumaric acid, and flavonoids such as rutin and kaempferol 3-*O*-glucoside were found in field pea. Caprioli et al. [28] reported much lower levels (<2.7 mg/kg DW) of gallic acid, coumaric acid, ferulic acid, catechin and kaempferol 3-*O*-glucoside n pea (*Pisum sativum*) compared to our data, while Magalhães et al. [30] found higher levels of *p*-hydroxybenzoic acid in different field pea cultivars (45–102 mg/kg DW).

Table 3 Quinic acid levels ranged from 407 μg/g DW in the faba bean cultivar Zoran to 844 μg/g DW in white lupin. Baicalein was found in significant concentrations (>341 μg/g DW) in white and blue lupin, faba beans and red lentil. Isorhamnetin was found in white lupin (124 μg/g DW) and the faba bean Zoran (129 μg/g DW), while luteolin 7-*O*-glucoside was found mainly in yellow lupin (129 μg/g DW). Catechin was found in faba beans and brown lentil, while epicatechin was found only in faba beans (>575 μg/g DW). Magalhães et al. [30] reported higher levels of *p*-hydroxybenzoic acid in different chickpea cultivars (19–41 mg/kg DW). Significant levels of catechin and epicatechin were previously detected in faba beans and lentils [29]. Higher levels of gallic acid (23–138 mg/kg DW), much lower levels of epicatechin (0–222 mg/kg DW), and no *p*-hydroxybenzoic acid compared to our data were found in several faba bean cultivars [30].

The total phenolic content in the seeds of the studied pulses showed great differences, as it ranged from 719 μg/g DW to 5012 μg/g DW (Figure 1). Among the pulses, the total phenolic content averaged among all cultivars and accessions was highest in common bean, followed by faba bean and white lupin. Pulses with total phenolic content above 1000 μg/g DW were common and runner bean, white and blue lupin, faba bean and red lentil. Pulses with a total phenolic content below 1000 μg/g DW were chickpea, brown lentil, field pea and yellow lupin. Kumar et al. [19] reported higher TPCs for chickpeas, field peas and faba beans. 

Flavonoids and phenolic acids were the most represented class of all phenolic compounds in these pulse samples (Figure 2a). Total phenolic acids ranged from 3 to 102 μg/g DW, and flavonoids ranged from 0 to 4063 μg/g DW. The relative distribution of total phenolic acids compared with flavonoids was highest in chickpea (100%), followed by brown lentil (47%), runner bean (28%), red lentil (26%) and field pea (19%). On the other hand, the relative distribution of total flavonoids compared to phenolic acids was highest in white lupin and the four common bean samples (99%). Blandino et al. [32] reported lower flavonoid contents for red lentil and green pea, 68.1 mg/kg DW and 49.7 mg/kg DW, respectively. However, they determined a relatively high content of flavonoids in chickpea (10.7 mg/kg DW), whereas no flavonoids (<LOQ) were detected in our chickpea sample. The amount of phenolic compounds in pulses and grain legumes depends on the variety, climatic conditions and processing, but their relative distribution is generally in favour of flavonoids, which was also confirmed here, although it is claimed that they generally contain about 60% flavonoids and 40% phenolic acids [14]. Similar results were presented by Blandino et al. [32], who found a high relative content of hydroxybenzoic acid in chickpeas, while green peas contained mainly hydroxycinnamic acids.

Further phenolic acid distribution analysis showed that more hydroxybenzoic acids than hydroxycinnamic acids were present (≥55%) in one common bean (SRGB204), runner bean, faba beans, red and brown lentil and chickpea, expressed as a ratio of total phenolic content calculated from our results (Figure 2b). Pulses with less hydroxybenzoic acids than hydroxycinnamic acids (≤8%) were one common bean (SRGB196) and field pea, expressed relative to total phenolic content. Four common beans (SRGB Škobrne, SRGB304, Etna and Golden gate) and white, blue and yellow lupins did not contain hydroxybenzoic acids.

Further distribution analysis of flavonoids showed that more flavones than flavonols and flavanols (≥72%) were present in one common bean (SRGB204) and white, blue and yellow lupins, expressed as a ratio of total phenolic content (Figure 2c). A higher content of flavonols than of flavones and flavanols (68%) was found only in one sample of common bean (SRGB Škobrne). Pulses with more flavanols than flavones and flavonols (≤49%) were two common beans (Etna and Golden gate), runner bean, field pea, faba beans and red lentil, expressed relative to total phenolic content. Among the pulses studied, three common beans (Etna, Golden gate and SRGB196), runner bean, field pea and chickpea did not contain flavones, one common bean (Golden gate) and red lentil did not have flavonols, and flavanols were not present in the two common beans (SRGB204 and SRGB196), white, blue and yellow lupins, and red lentil. Chickpeas did not contain flavonoids, hence the blank distribution histogram in Figure 3c. Blandino et al. [32] reported lower total phenolic acid and flavonoid contents in chickpea, red lentil and field pea. The differences in phenolic classes in pulses can be attributed to various factors such as genotype, growing conditions, and storage [29,33]. It is often pointed out that genotype has the greatest influence on phenolic compound content in pulses and grain legumes [30]. Therefore, the importance of screening different genotypes is more important than maturity and environmental conditions [30].

### 2.3. Multi-Mineral Profile

The macro- and micro-mineral composition of sixteen pulse samples, determined by ICP-MS, is shown in Table 4. A total of ten minerals were determined and divided into two groups: the macro-minerals (>1 g/kg DW) Mg, P, S, K and Ca, and micro-minerals (>1 mg/kg DW) Cr, Mn, Fe, Zn and Mo (Table 4). The order of minerals from most abundant to least abundant (based on the mean values) is K (13.10 g/kg DW) > P (5.06 g/kg DW) > S (2.19 g/kg DW) > Mg (1.52 g/kg DW) > Ca (1.19 g/kg DW) for the macro- and Fe (55.17 mg/kg DW) > Zn (33.43 mg/kg DW) > Mn (25.54 mg/kg DW) > Mo (2.70 mg/kg DW) > Cr (0.52 mg/kg DW) for the micro-minerals in the analysed pulses. The essential elements for which the European Food Safety Authority (EFSA) [34] has established dietary reference values are Mg, Ca, P, K, Fe, Zn, Mn and Mo.

The ranges of macro-minerals were as follows: K (7.75–19.05 g/kg DW), P (2.94–7.47 g/kg DW), S (1.48–4.69 g/kg DW), Mg (0.76–2.93 g/kg DW) and Ca (0.24–2.27 g/kg DW). The calculated coefficient of variability was highest for Ca (40.41%), followed by S (33.48%) and Mg (30.51%). Among the pulses, the highest K concentration was found in the common bean sample and the lowest in brown lentils. Pulses with K content higher than 15 g/kg DW were three common bean samples and runner bean, and with K content lower than 10 g/kg DW were field pea, red lentil and brown lentil. P concentration was highest in yellow lupin and lowest in chickpea. Pulses with P content above 6 g/kg DW were white and yellow lupin and faba bean, and with P content below 3 g/kg DW were chickpea. The S concentration was highest in yellow lupin and lowest in a faba bean sample. Pulses with S content greater than 2 g/kg DW were five common bean samples, white, blue and yellow lupin, and chickpea, and with S content less than 2 g/kg DW were one common bean sample, runner bean, field pea, faba bean, red lentil and brown lentil. Mg concentration was highest in yellow lupin and lowest in red lentil. Pulses with Mg content greater than 2 g/kg DW were yellow lupin, and with Mg content, less than 1 g/kg DW were red and brown lentil. Ca concentration was highest in blue lupin and lowest in red lentil. Pulses with Ca content above 2 g/kg DW were blue lupin and with Ca content below 1 g/kg DW were three common bean samples, field pea, red and brown lentil. Here, the largest relative difference was observed, as the blue lupin contained a 10-fold higher Ca concentration than the red lentil.

The ranges of micro-minerals in pulse species studied were as follows: Fe (30.39–77.34 mg/kg DW), Zn (19.62–63.27 mg/kg DW), Mn (8.34–82.92 mg/kg DW), Mo (0.54–8.46 mg/kg DW) and Cr (0.15–2.88 mg/kg DW). Among them, the highest coefficient of variability was calculated for Cr (137.57%), followed by Mn (100.86%) and Mo (82.74%). Among the pulses, the highest Fe concentration was found in one common bean sample and the lowest in white lupin. Pulses with Fe content above 50 mg/kg DW were common and runner bean, yellow lupin, red lentil and chickpea, and with Fe content below 50 mg/kg DW were field pea, white and blue lupin, faba bean and brown lentil. Zn content was highest in yellow lupin and lowest in brown lentil. The pulses with Zn content above 50 mg/kg DW were white and yellow lupin, while the pulses with Zn content below 20 mg/kg DW were brown lentil. Mn concentration was highest in yellow lupin and lowest in one common bean sample. The pulses with Mn content above 70 mg/kg DW were white, blue and yellow lupins, while the pulses with Mn content below 10 mg/kg DW were one common bean sample, field pea and brown lentil. Lupins (white, blue, and yellow) were a significantly higher source of micro-mineral Mn than other pulses. Mo concentration was highest in one common bean sample and lowest in brown lentil. Pulses with Mo content greater than 5 mg/kg DW were two common bean samples and red lentil, and with Mo content less than 1 mg/kg DW were two common bean samples, field pea, faba bean and brown lentil. Cr concentration was highest in yellow lupin and lowest in red lentils. Pulses with Cr content above 1.5 mg/kg DW were blue and yellow lupins.

This multi-mineral composition is consistent with data reported for common and runner beans [35,36,37,38], field peas [39], lupins [40], faba beans [40,41], lentils [39,42,43] and chickpeas [42,44]. Guild et al. [37] reported similar Fe contents for common bean (68.4 mg/kg DW), while the average Zn content was slightly higher (31.1 mg/kg DW). Oliveira et al. [36] studied seven *Phaseolus* bean varieties from the local market in Brazil, whose contents of Ca, S, Fe and Zn were comparable to our data. However, the range of contents among bean varieties was wider than in our study, e.g., Fe content ranged from 56–96 mg/kg DW and Zn content ranged from 33–58 mg/kg DW. Alvarado-López et al. [38] studied four runner bean varieties (*Phaseolus coccineous*) with different seed colours, i.e., black, purple, white and brown, and found lower K and Mg contents, 12.4–14.4 g/kg DW and 1.5 g/kg DW, respectively. Ciurescu et al. [39] reported higher or comparable contents for all analysed minerals (Ca, P, Fe, Mn and Zn) in five pea and four lentil cultivars. Greater differences in contents were found between cultivars than between species (pea vs. lentil). All analysed macro-minerals in lupins were comparable to the contents from the study by Lizarazo et al. [40]. However, slightly lower values were found for Mg (1.8 g/kg DW) and P (3.6 g/kg DW). Faba beans were also analysed in the same study. The results obtained are comparable to our data; only Ca and Mn values are slightly lower, 0.8 g/kg DW and 8.8 mg/kg DW, respectively. The levels of K, Ca, Mg and Fe in chickpea were higher in the study of Ereifej et al. [44], while they were lower for these minerals in the study of Farooq et al. [42].

### 2.4. Cluster and Multivariate Analysis

Statistical analysis of the results was performed using cluster analysis (Ward’s method and squared Euclidean distances) to determine the distribution of pulses according to the nutritional characteristics studied. The dendrogram in Figure 3 shows the formation of three main groups. The first group included seven samples of *Phaseolus vulgaris*—common bean (SRGB Škobrne, SRGB204, SRGB304, Etna, Golden gate and SRGB196) and *Phaseolus coccineus*—runner bean (SRGB222); the second group included eight samples of *Pisum sativum*—field pea (Eso), *Lupinus albus*—white lupin (Energy), *Lupinus angustifolius*—blue lupin (Sonet), *Vicia faba*—faba bean (Zoran, Merkur), *Lens culinaris*—lentil (brown, red)*,* and *Cicer aretinum*—chickpea; and the third group included only *Lupinus luteus*—yellow lupin (Mister).

The matrix of 40 variables for the 16 pulse samples was evaluated using Pearson’s rank correlation coefficients to a significance of *p* < 0.05 (Figure 4). Crude protein content showed very strong significant correlations with Zn and Mn (>0.82), and strong significant correlations with S, luteolin 7-*O*-glucoside and apigenin 7-*O*-glucoside (>0.67). Crude fat content had a very strong significant correlation with Mn (0.84) and a strong significant correlation with Ca (0.77). A moderate correlation was reported previously for protein with Zn and fat with Mn in field peas [45]. Similarly, for lentils, crude protein content showed a high correlation with Zn and Mn, and fat with Mn and Ca [46].

A perfect positive significant correlation (1.00) was observed for the following pairs of variables: apigenin—genistein, chrysoeriol—genistein, luteolin 7-*O*-glucoside—genistein, chrysoeriol—apigenin, luteolin 7-*O*-glucoside—apigenin and luteolin 7-*O*-glucoside—chrysoeriol. *p*-Coumaric acid showed very strong significant correlations with hyperoside and quercitrin (>0.91), while ferulic acid showed a strong significant correlation with quinic acid (0.62) and quinic acid with K (0.70). Genistein showed very strong significant correlations with luteolin 7-*O*-glucoside, S and Cr (>0.84), and strong significant correlations with Mg and Zn (>0.64). Ametoflavone showed a very strong significant correlation with isorhamnetin (0.91) and a strong significant correlation with baicalein, TPC, and rutin (>0.64). Apigenin showed very strong significant correlations with apigenin 7-*O*-glucoside, S and Cr (>0.84), and strong significant correlations with Mg and Zn (>0.64). Apigenin 7-*O*-glucoside showed very strong significant correlations with luteolin 7-*O*-glucoside, chrysoeriol, S and Cr (>0.82), and strong significant correlations with Mg and Zn (>0.68). Apiin showed a very strong significant correlation with vitexin (0.85) and baicalein with isorhamnetin (0.85). Chrysoeriol and luteolin 7-*O*-glucoside showed very strong significant correlations with S and Cr (>0.84), and strong significant correlations with Mg and Zn (>0.64). Vitexin showed strong significant correlations with Ca, Cr and Mg (>0.67), and isorhamnetin with TPC and rutin (>0.60). Hyperoxide had a very strong significant correlation with quercitrin (0.95) and kaempferol with kaempferol 3-*O*-glucoside (0.98). In addition, kaempferol 3-*O*-glucoside showed strong significant correlations with quercetin 3-*O*-glucoside (0.66). Rutin showed a strong significant correlation with TPC (0.63). Quercetin showed a very strong significant correlation with quercetin 3-*O*-glucoside and TPC (>0.88) and quercetin 3-*O*-glucoside with TPC (0.89).

Among minerals, Mg showed a very strong significant correlation with S (0.84) and strong significant correlations with P and Cr (>0.69). Moderate correlations between P and Mg have also been reported for field peas [45]. P showed a strong significant correlation with Zn (0.69) and S with Cr, Mn and Zn (>0.61). Ca showed a very strong significant correlation with Mn (0.86), and a strong significant correlation with Cr (0.62). Finally, Cr showed a strong significant correlation with Mn (0.78) and Mn with Zn (0.71). The highest negative (moderately strong) correlation was calculated for Ca with *p*-hydroxybenzoic acid (–0.57).

Statistical evaluation of these data was performed by multiple comparisons to reveal variations within pulses according to specific nutritional characteristics. This distribution was visualized as a Box and Whisker plot, which can be seen in Figure 5. Overall, phytic acid and TPC showed the highest response in all 16 pulse samples analysed. In addition, quinic acid, baicalein and catechin showed a moderate response, followed by isorhamnetin, quercetin 3-*O*-glucoside and epicatechin. For the nutrients analysed (crude protein, crude fat and macro/micro-minerals), the responses were low when comparing the different pulses. Phytic acid, as an antinutrient, was one of the most deviating parameters among the pulses studied. Therefore, it represents the greatest potential for the improvement of different pulse species in breeding procedures. According to this available information, the genetic resources/different species of pulses with the lowest phytic acid content could be used and included for further selection. Among the bioactive compounds, quinic acid, baicalein and catechin showed the highest response, which can also be used in breeding programmes. In particular, TPC reflects the highest improvement ability in pulse breeding.

## 3. Materials and Methods

### 3.1. Plant Material

A set of sixteen homogenised marketable seeds of ten pulse species was obtained for the nutritional analyses (Table 1). The following samples were analysed: six common beans (*Phaseolus vulgaris* L.), one runner bean (*Phaseolus coccineus* L.), one field pea (*Pisum sativum* L. subsp. *arvense*), one white lupin (*Lupinus albus* L.), one blue lupin (*Lupinus angustifolius* L.), one yellow lupin (*Lupinus luteus* L.), two faba bean (*Vicia faba* L. var. minor), two lentils (*Lens culinaris* L.) and one chickpea (*Cicer aretinum* L.) sample. Most of these pulses were grown in the experimental fields of Infrastructure Centre Jablje, Agricultural Institute of Slovenia (304 m a.s.l.; 46.151° N 14.562° E) according to the production techniques established for each species. Three samples of pulses (brown lentil, red lentil and chickpea) were purchased in the food retail market, so the cultivar is unknown (Table 1). Representative samples of the air-dried seeds were pulverised and homogenised using a laboratory ball mill (Retsch MM 400; Retsch GmbH, Haan, Germany) at a high frequency of 30 Hz for 2–4 min before analysis.

### 3.2. Moisture, Protein and Fat Content

Moisture content was determined by drying the samples at 103 °C for 48 h. Crude protein was analysed by the Kjeldahl method (ISO 5983:2) with a factor of 6.25. The method consists of three consecutive steps, i.e., acid digestion with sulfuric acid, distillation with alkali (NaOH) and titration with an HCl standard solution in the presence of an indicator. The results are expressed in %. The crude fat was analysed based on extraction with petroleum ether. Five grams of the homogenised sample was placed in an extraction quiver, placed in an extractor and extracted with petroleum ether for six hours. The petroleum extract was collected in a dry-weighed flask. The solvent was distilled off and the residue was dried in a drying oven to constant weight. The results are expressed in % per dry weight (DW).

### 3.3. Phytic Acid

Phytic acid content was determined according to the modified method of Haug and Lantzsch [47]. It was based on the indirect spectrophotometric determination of phytic phosphorus in dry bean extracts. Phytic acid was precipitated by the addition of ferric ammonium sulphate. Some of the iron formed insoluble ferric phytate, and the remaining iron was determined spectrophotometrically at 519 nm with 2,2′-bipyridine. A calibration curve was established by a series of standard solutions of a sodium salt of phytic acid. Half a gram of the homogenised sample was extracted with 100 mL of 2.4% HCl for 3 h with constant stirring. The extract was filtered through Whatman filter paper No. 41 and 0.5 mL of the extract was transferred to a stoppered glass tube; ammonium iron (III) sulfate solution (0.2 g NH_4_Fe(SO_4_)_2_ × 12 H_2_O dissolved in 100 mL 2 mol/L HCl and filled to the mark with distilled water) was added. A closed glass tube was held in a boiling water bath for 30 min, then cooled in an ice bath for 15 min and left to attain room temperature. The tube was centrifuged at 3000 rpm and 1 mL of the supernatant was transferred to another glass tube and 1.5 mL of 2,2′-bipyridine solution (10 g of 2,2′-bipyridine dissolved in 10 mL of thioglycolic acid and filled to the mark with distilled water) was added. The absorbance was measured at 519 nm [26]. The results are expressed in mg/100 g DW.

### 3.4. HPLC-MS/MS Analysis

The extraction of homogenised samples was performed according to the method developed by Šibul et al. [48,49]. Extracts were prepared by maceration of powdered and homogenised air-dried samples with 80% aqueous MeOH (13 mL per 1 g of material) for 90 min, with continuous shaking at room temperature. The seed material was removed by filtration and re-extracted with three additional batches of the fresh solvent. The crude extracts were pooled and evaporated under reduced pressure and reconstituted in DMSO. For HPLC-MS/MS determination of phenolic profile, a method for quantification of 25 phenolic compounds commonly found in plants was used as follows: All extracts were diluted with mobile phase A (0.05% aqueous formic acid) and B (methanol) solvents and premixed at a ratio of *1:1* to obtain a final concentration of 2 mg/mL. Fifteen working standards ranging from 1.53 ng/mL to 25.0 × 10^3^ ng/mL were prepared by serial *1:1* dilution of the standard mixture with solvents A and B (*1:1*). Samples and standards were analysed using an Agilent Technologies 1200 Series high-performance liquid chromatograph coupled to an Agilent Technologies 6410A Triple Quad tandem mass spectrometer with an electrospray ion source and controlled by Agilent Technologies MassHunter Workstation Software—Data Acquisition (vB.03.01; Agilent Technologies, Inc., Santa Clara, CA, USA). Five μL was injected into the system and compounds were separated on a Zorbax Eclipse XDB-C18 (50 mm × 4.6 mm, 1.8 µm) fast resolution column at 50 °C. The mobile phase was eluted at a flow rate of 1 mL/min in a gradient mode (0 min 30% B, 6 min 70% B, 9 min 100% B, 12 min 100% B and re-equilibration time 3 min). The eluted compounds were detected using ESI-MS. The following ion source parameters were used: nebulisation gas pressure (N_2_) 40 psi, drying gas flow rate (N_2_) 9 L/min and temperature 350 °C, capillary voltage 4 kV and negative polarity. Data were recorded in dynamic MRM mode using the optimised compound-specific parameters (retention time, precursor ion, product ion, fragmented voltage and collision voltage; Table 5) from the previously published study by Orčić et al. [31]. Peak areas were determined for all compounds using Agilent MassHunter Workstation Software—Qualitative Analysis (vB.03.01). The reference standards of the phenolic compounds were purchased from Sigma–Aldrich Chem (Steinheim, Germany), Fluka Chemie GmbH (Buchs, Switzerland), and Chromadex (Santa Ana, CA, USA). Calibration curves were generated and sample concentrations were calculated using OriginLabs Origin Pro software (v8.0 OriginLab Corporation, Northampton, MA, USA) [48]. The results are given in μg/g DW.

### 3.5. ICP-MS Analysis

Multi-mineral analysis (Mg, P, S, K, Ca, Cr, Mn, Fe, Zn and Mo) was performed using inductively coupled plasma mass spectrometry (ICP-MS). Homogenised samples (250 mg) were mixed with 6 mL nitric acid (65%, *v/v*; Suprapur, Merck) and 2 mL of hydrogen peroxide (30%, *v/v*; Suprapur, Merck) and digested using the Ethos UP microwave digestion system. The digested solutions were diluted to 50 mL with 2× deionized water. Elements in the samples were determined using an Agilent ICP-MS 7900 (Tokyo, Japan) with the octopole reaction system. Helium (He) was used as the reaction gas with a flow rate of 5 mL/min in He mode and 10 mL/min in HEHe mode. The calibration curve was prepared using IV-STOCK-50 standard solution (Inorganic Ventures, Christiansburg, VA, USA) and single standard solutions of P and S (Inorganic Ventures, USA) were added separately to the mixture. A certified reference material (NIST SRM 1573a tomato leaves, Gaithersburg, MD, USA) was used to verify the accuracy of the results. All results are reported as g/kg DW for macro-minerals or mg/kg DW for micro-minerals.

### 3.6. Statistical Analysis

Statistical calculations and multivariate analysis were performed using the program Statgraphics Centurion v18.1.16 (StatPoint Technologies, Inc., Warrenton, VA, USA). Cluster analysis of the different pulses was performed using Ward’s method with squared Euclidean distance metric and data standardisation. Pairwise associations between individual nutrients, phytic acid, and bioactive compounds were evaluated using Pearson correlation analysis (1.0 = |r|, perfect correlation; 0.8 < |r| < 1.0, very strong correlation; 0.6 < |r| < 0.8, strong correlation; 0.4 < |r| < 0.6, moderately strong correlation). Multiple sample comparisons were calculated to show variation within studied pulse species by specific nutritional characteristics. This distribution was visualized as a Box and Whisker plot.

## 4. Conclusions

This study provides an assessment of nutrients (protein, fat and macro-/micro-minerals), antinutrients (phytic acid) and several bioactive compounds belonging to phenolic acids and/or flavonoids in ten pulse species. Significant differences in the contents of all studied compounds were confirmed between pulse species and among different cultivars/genotypes. The greatest differences were found in the content of phytic acid and twenty-five phenolic compounds. The total phenolic content and the relative phenolic classes varied considerably among all studied types of pulses. Because high variability in analysed (anti)nutritional compounds was found among the pulses studied, there is potential for targeted cultivation and use of specific pulse species in agro-food systems.

## Figures and Tables

**Figure 1 plants-12-00170-f001:**
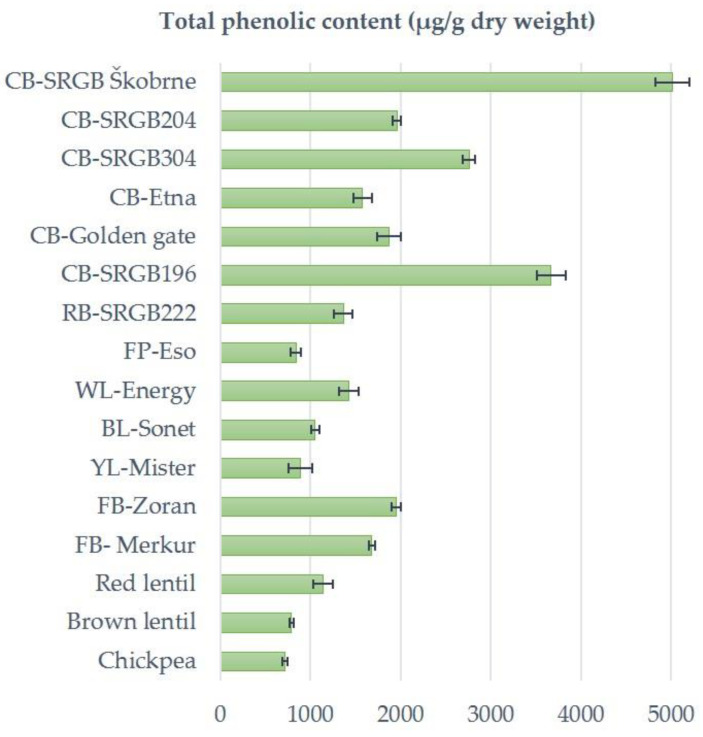
Total phenolic content in seeds of studied pulses. Data are means (±standard deviation) of three replicates. CB, common bean; RB, runner bean; FP, field pea; WL, white lupin; BL, blue lupin; YL, yellow lupin; FB, faba bean.

**Figure 2 plants-12-00170-f002:**
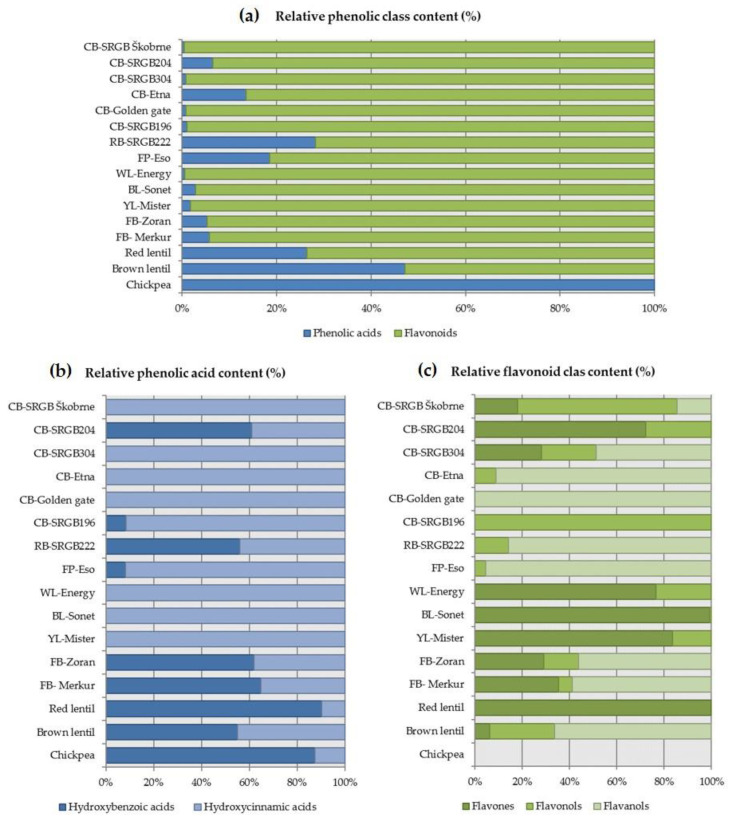
Principal qualitative profiles of phenolic classes (**a**), phenolic acid classes (**b**), and flavonoid classes (**c**) in seeds of 16 pulse samples. CB, common bean; RB, runner bean; FP, field pea; WL, white lupin; BL, blue lupin; YL, yellow lupin; FB, faba bean.

**Figure 3 plants-12-00170-f003:**
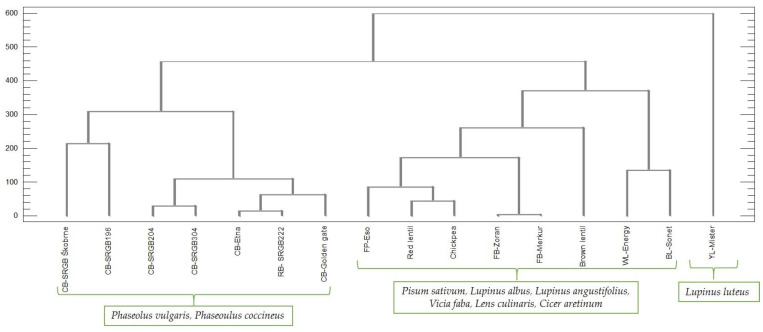
Dendrogram (Ward’s method, squared Euclidean distances) for 16 pulse samples according to 40 variables. CB, common bean; RB, runner bean; FP, field pea; WL, white lupin; BL, blue lupin; YL, yellow lupin; FB, faba bean.

**Figure 4 plants-12-00170-f004:**
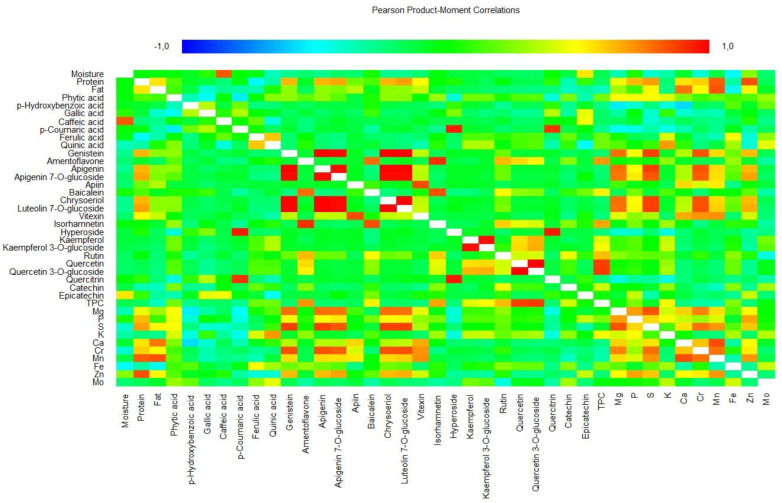
Pearson’s correlation matrix for 40 variables studied in pulse samples. TPC, total phenolic compounds.

**Figure 5 plants-12-00170-f005:**
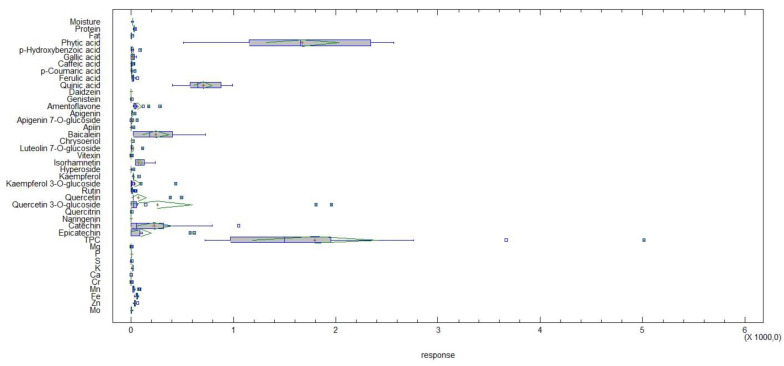
Box and Whisker Plot for 40 variables studied in pulse samples.

**Table 1 plants-12-00170-t001:** Nutritional properties and phytic acid content in the seeds of studied pulses.

Species	Common Name	Code/ Cultivar Name	Moisture (%)	Crude Protein (%)	Crude Fat (%)	Phytic Acid (mg/100 g DW)
*Phaseolus vulgaris* L.	Common bean	SRGB Škobrne	9.0	23.9	1.0	1745
		SRGB204	8.9	24.0	0.6	2566
		SRGB304	9.1	22.1	1.4	1743
		Etna	11.2	20.2	1.3	1401
		Golden gate	9.1	24.3	0.8	2550
		SRGB196	9.4	23.0	0.9	2206
*Phaseolus coccineus* L.	Runner bean	SRGB222	8.1	18.0	2.0	1115
*Pisum sativum* L. subsp. *arvense*	Field pea	Eso	15.6	18.5	1.6	2399
*Lupinus albus* L.	White lupin	Energy	13.5	38.4	8.5	2315
*Lupinus angustifolius* L.	Blue lupin	Sonet	8.4	31.7	5.4	1386
*Lupinus luteus* L.	Yellow lupin	Mister	7.8	43.1	4.8	2368
*Vicia faba* L. var. minor	Faba bean	Zoran	13.6	28.3	0.9	1564
		Merkur	14.4	29.4	0.8	507
*Lens culinaris* L.	Red lentil	unknown *	9.7	25.9	1.2	1191
	Brown lentil	unknown *	9.4	27.6	0.8	612
*Cicer aretinum* L.	Chickpea	unknown *	9.2	19.1	5.4	1116
Range			7.8–15.6	16.7–43.1	0.6–8.5	507–2566

* purchased in food retail market; SRGB, Slovenian plant gene bank; DW, dry weight.

**Table 2 plants-12-00170-t002:** Quantification of phenolic compounds (μg/g dry weight (DW)) in common bean, runner bean and field pea seeds *.

Compound	Common Bean						Runner Bean	Field Pea
	SRGB Škobrne	SRGB204	SRGB304	Etna	Golden Gate	SRGB196	SRGB222	Eso
*p*-Hydroxybenzoic acid	<0.16 **	2.98 (0.18)	<0.16	<0.16	<0.16	2.54 (0.15)	5.88 (0.35)	4.21 (0.25)
Gallic acid	<0.12	31.8 (2.9)	<0.12	<0.12	<0.12	<0.12	49.5 (4.5)	<0.12
Caffeic acid	<0.8	<0.8	<0.8	10.2 (0.7)	<0.8	<0.8	<0.8	24.8 (1.7)
*p*-Coumaric acid	1.38 (0.12)	1.63 (0.15)	2.22 (0.20)	10.90 (0.98)	1.09 (0.10)	3.17 (0.29)	5.10 (0.46)	4.86 (0.44)
Ferulic acid	19.80 (0.02)	20.50 (0.02)	12.50 (0.01)	65.40 (0.07)	7.95 (0.01)	24.40 (0.02)	38.50 (0.04)	16.80 (0.02)
Quinic acid	646 (0.7)	922 (0.9)	767 (0.8)	917 (0.9)	709 (0.7)	916 (0.9)	992 (1.0)	545 (0.6)
Genistein	<0.2	<0.2	<0.2	<0.2	<0.2	<0.2	<0.2	<0.2
Amentoflavone	282 (8.5)	171 (5.1)	118 (3.5)	20.5 (0.6)	19.8 (0.6)	18.1 (0.5)	21.6 (0.7)	20.7 (0.6)
Apigenin	<8	<8	<8	<8	<8	<8	<8	<8
Apigenin 7-*O*-glucoside	<0.2	<0.2	<0.2	<0.2	<0.2	<0.2	<0.2	<0.2
Apiin	<0.06	<0.06	<0.06	<0.06	<0.06	<0.06	<0.06	<0.06
Baicalein	724 (2.2)	579 (1.7)	518 (1.6)	<16	<16	<16	<16	<16
Chrysoeriol	<4	<4	<4	<4	<4	<4	<4	<4
Luteolin 7-*O*-glucoside	5.94 (0.18)	4.96 (0.15)	4.95 (0.15)	<0.2	<0.2	<0.2	<0.2	<0.2
Vitexin	<0.2	<0.2	<0.2	<0.2	<0.2	<0.2	<0.2	<0.2
Isorhamnetin	236 (14)	174 (10)	149 (9)	<40	<40	<40	<40	<40
Hyperoside	<0.06	<0.06	<0.06	<0.06	<0.06	<0.06	<0.06	<0.06
Kaempferol	<16	<16	<16	<16	<16	77.2 (5.4)	<16	<16
Kaempferol 3-*O*-glucoside	16.2 (0.7)	<0.08	95.9 (3.8)	8.62 (0.34)	<0.08	436 (17)	<0.08	2.85 (0.11)
Rutin	40.1 (1.2)	6.15 (0.18)	46.3 (1.4)	9.38 (0.28)	<2	7.82 (0.23)	10.90 (0.33)	6.87 (0.21)
Quercetin	494 (1.5)	<16	<16	<16	<16	379 (1.1)	<16	<16
Quercetin 3-*O*-glucoside	1958 (59)	44.6 (1.3)	140 (4.2)	31.0 (0.9)	<0.06	1804 (54)	24.3 (0.7)	<0.06
Quercitrin	<0.06	<0.06	<0.06	<0.06	<0.06	<0.06	<0.06	<0.06
Catechin	589 (6)	<0.4	793 (8)	421 (4)	1052 (11)	<0.4	216 (2)	212 (2)
Epicatechin	<0.4	<0.4	112.8 (1.3)	83.4 (0.8)	81.4 (0.8)	<0.4	<0.4	<0.4

* Data are mean (standard deviation) of three replicates from one seed batch. ** Compounds below quantification limit were given as <LoQ, where LoQ is method quantification limit, calculated from instrument quantification limit (given in Orčić et al. [31]) and sample dilution.

**Table 3 plants-12-00170-t003:** Quantification of phenolic compounds (μg/g dry weight (DW)) in lupins (white, blue and yellow), faba bean, lentil and chickpea seeds *.

Compound	White Lupin	Blue Lupin	Yellow Lupin	Faba Bean		Red Lentil	Brown Lentil	Chickpea
	Energy	Sonet	Mister	Zoran	Merkur			
*p*-Hydroxybenzoic acid	<0.16 **	<0.16	<0.16	8.65 (0.52)	7.55 (0.45)	88.3 (5.3)	17.7 (1.1)	18.4 (1.1)
Gallic acid	<0.12	<0.12	<0.12	41.1 (3.7)	34.9 (3.1)	31.4 (2.8)	31.4 (2.8)	<0.12
Caffeic acid	<0.8	<0.8	<0.8	15.0 (1.1)	10.9 (0.8)	<0.8	<0.8	<0.8
*p*-Coumaric acid	<0.2	1.72 (0.15)	2.55 (0.23)	4.77 (0.43)	6.00 (0.54)	7.76 (0.70)	35.80 (3.22)	<0.2
Ferulic acid	2.88 (0.003)	9.31 (0.009)	2.61 (0.003)	10.60 (0.01)	6.04 (0.006)	4.87 (0.005)	4.14 (0.004)	2.61 (0.003)
Quinic acid	844 (0.8)	628 (0.6)	569 (0.6)	407 (0.4)	525 (0.5)	610 (0.6)	574 (0.6)	653 (0.7)
Genistein	<0.2	<0.2	4.14 (0.29)	<0.2	<0.2	<0.2	<0.2	<0.2
Amentoflavone	50.6 (1.5)	36.8 (1.1)	34.1 (1.0)	45.4 (1.4)	40.3 (1.2)	29.5 (0.9)	27.9 (0.8)	45.4 (1.4)
Apigenin	<8	<8	35.9 (2.5)	<8	<8	<8	<8	<8
Apigenin 7-*O*-glucoside	<0.2	1.51 (0.08)	59.6 (3.0)	<0.2	13.0 (0.7)	<0.2	<0.2	<0.2
Apiin	<0.06	20.1 (1.0)	<0.06	<0.06	<0.06	<0.06	<0.06	<0.06
Baicalein	399 (1.2)	341 (1.0)	<16	405 (1.2)	354 (1.1)	368 (1.1)	<16	<16
Chrysoeriol	<4	<4	17.2 (0.5)	<4	<4	<4	<4	<4
Luteolin 7-*O*-glucoside	4.95 (0.15)	5.57 (0.17)	113 (3.4)	4.88 (0.15)	4.76 (0.14)	<0.2	6.03 (0.18)	<0.2
Vitexin	<0.2	5.25 (0.26)	2.70 (0.14)	1.88 (0.09)	<0.2	<0.2	<0.2	<0.2
Isorhamnetin	124 (7)	<40	<40	129 (8)	<40	<40	<40	<40
Hyperoside	<0.06	<0.06	<0.06	<0.06	<0.06	<0.06	20.4 (1.2)	<0.06
Kaempferol	<16	<16	<16	<16	<16	<16	<16	<16
Kaempferol 3-*O*-glucoside	<0.08	<0.08	<0.08	2.15 (0.09)	1.54 (0.06)	<0.08	<0.08	<0.08
Rutin	<2	3.38 (0.10)	22.1 (0.7)	10.3 (0.3)	14.3 (0.4)	<2	<2	<2
Quercetin	<16	<16	<16	<16	<16	<16	<16	<16
Quercetin 3-*O*-glucoside	<0.06	<0.06	23.4 (0.7)	65.0 (2.0)	45.0 (1.4)	<0.06	<0.06	<0.06
Quercitrin	<0.06	<0.06	<0.06	2.32 (0.14)	<0.06	<0.06	7.04 (0.42)	<0.06
Catechin	<0.4	<0.4	<0.4	184 (2)	43.5 (0.4)	<0.4	66.1 (0.7)	<0.4
Epicatechin	<0.4	<0.4	<0.4	613 (6)	575 (6)	<0.4	<0.4	<0.4

* Data are mean (standard deviation) of three replicates from one seed batch. ** Compounds below quantification limit were given as <LoQ, where LoQ is method quantification limit, calculated from instrument quantification limit (given in Orčić et al. [31]) and sample dilution.

**Table 4 plants-12-00170-t004:** Multi-mineral composition in the seeds of 16 pulse samples.

Common Name	Code/Cultivar Name	Macro-Minerals (g/kg DW)					Micro-Minerals (mg/kg DW)				
		Mg	P	S	K	Ca	Cr	Mn	Fe	Zn	Mo
Common bean	SRGB Škobrne	1.42	4.82	2.12	13.93	0.99	0.26	11.95	64.59	28.06	0.57
	SRGB204	1.64	5.90	2.28	19.05	1.18	0.18	12.35	69.91	27.94	3.19
	SRGB304	1.55	5.00	2.16	14.41	0.94	0.33	10.97	57.75	28.93	0.85
	Etna	1.47	4.18	1.97	13.53	1.33	0.21	13.59	77.34	26.76	5.83
	Golden gate	1.67	5.87	2.38	16.25	1.18	0.19	11.86	58.74	27.63	8.46
	SRGB196	1.73	5.87	2.20	17.84	0.81	0.21	9.45	56.92	20.32	4.69
Runner bean	SRGB222	1.76	4.65	1.99	17.77	1.12	0.29	12.65	52.70	23.02	2.43
Field pea	Eso	1.30	3.66	1.58	9.11	0.91	0.23	8.34	46.84	26.35	0.56
White lupin	Energy	1.46	6.19	2.88	12.32	1.98	0.21	72.79	30.39	52.88	3.11
Blue lupin	Sonet	1.75	4.89	2.23	11.05	2.27	1.69	79.54	43.93	34.87	1.72
Yellow lupin	Mister	2.93	7.47	4.69	12.08	1.83	2.88	82.92	64.87	63.27	1.41
Faba bean	Zoran	1.36	6.46	1.48	12.55	1.07	0.18	16.41	43.27	49.13	0.92
	Merkur	1.40	6.19	1.64	12.67	1.02	0.16	16.79	46.88	44.55	0.97
Red lentil	not defined *	0.76	3.72	1.82	9.22	0.24	0.15	13.65	65.89	33.00	5.26
Brown lentil	not defined *	0.78	3.14	1.57	7.75	0.78	0.26	9.62	47.32	19.62	0.54
Chickpea	not defined *	1.33	2.94	2.05	10.08	1.31	0.96	25.77	55.46	28.57	2.74
Range		0.76–2.93	2.94–7.47	1.48–4.69	7.75–19.05	0.24–2.27	0.15–2.88	8.34–82.92	30.39–77.34	19.62–63.27	0.54–8.46

* purchased in food retail market; SRGB, Slovenian plant gene bank; DW, dry weight.

**Table 5 plants-12-00170-t005:** Optimised HPLC-MS/MS dynamic MRM parameters.

Compound	Precursor Ion (*m/z*)	Product Ion (*m/z*)	Retention Time (min)
*p*-Hydroxybenzoic acid	137	93	1.08
Gallic acid	169	125	0.58
Caffeic acid	179	135	1.18
*p*-Coumaric acid	163	119	1.69
Ferulic acid	193	134	1.90
Quinic acid	191	85	0.52
Genistein	269	133	4.12
Amentoflavone	537	375	5.78
Apigenin	269	117	4.71
Apigenin 7-*O*-glucoside	431	268	2.81
Apiin	563	269	2.60
Baicalein	445	269	3.40
Chrysoeriol	299	284	4.82
Luteolin 7-*O*-glucoside	447	285	2.13
Vitexin	431	311	1.90
Isorhamnetin	315	300	4.79
Hyperoside	463	300	2.16
Kaempferol	285	285	4.55
Kaempferol 3-*O*-glucoside	447	284	2.80
Rutin	609	300	2.33
Quercetin	301	151	3.74
Quercetin 3-*O*-glucoside	463	300	2.25
Quercitrin	447	300	2.75
Catechin	289	245	0.74
Epicatechin	289	245	0.95

## Data Availability

Not applicable.

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
