# Peer review of "Nutrients, Phytic Acid and Bioactive Compounds in Marketable Pulses"

_plants, 2022, doi:10.3390/plants12010170_

Round 1

Reviewer 1 Report

The authors presented a paper on Nutrients, phytic acid and bioactive compounds in marketable pulses.

The topic is interesting and may be within the aims and scopes of the Journal.

Yet, the manuscript needs several major implementations and changes especially in the Discussion section before it can be really considered for publication in this eminent Journal.

My comments are reported below one by one:

ABSTRACT:

- Line 14: Anti(nutrients)? What is this?

- Lines 15 on: Who did this? Is this your work? It is not clear.

- “The most varying phytochemicals in pulses were phytic acid, quinic acid, catechin, and TPC.” In what sense?

- Is this research new or not? Please specify it here.

INTRODUCTION:

- “They are defined as members of legumes belonging to the Leguminosae family...” Too intricate.

- Lines 44 on: It is not clear if this part regards all pulses or only your studied ones. Please specify for readers. If the former is true, your information is not always correct.

- Line 67-68: Phytic acid is not a negative molecule in nature. The bond with metals is via a coordination bond and not via an ion bond.

- Lines 69-71: Where does this fact of the galactose bonds comes from? You must specify earlier that verbascose is a polymer of α-galactose.

- Line 72: And what are all these anti-nutritional factors?

- Line 81: Why is phytic acid an antinutritional factor? This must be explained here.

- How did you get to choose right these seeds? What are the reasons for these choices?

- Is this whole research new or not? This aspect must be highlighted.

- What are the specific aims, scopes and final objectives of this research? This aspect must be highlighted.

RESULTS AND DISCUSSION:

- Lines 85-86: What is the sense of this?  

- This section must start with some information of the studied seeds for example, their provenience, their morphological characters, their organoleptic properties. You must highlight that you also studied more cultivars, and you must specify the environmental characteristics of these cultivars.

- You must specify if all these values in Table 1 are typical for your seeds and the reasons why the results are as such. In addition, you must give the general values of the comparisons. Lastly, you must specify if you compared our results with others from the same collection area or not. If not, you must divide these results and highlight them better. Most of all, you must specify if all these results are good or not from the nutraceutical point of view. A discussion on this last point is compulsory.

- Lines 121-132: This part is not for this section but for the introduction eventually.

- Line 127: Bound form?

- Lines 130-131: No way.

- “Compounds below the quantification limit were given as < LoQ, where LoQ is the method quantification limit, calculated from the instrument quantification limit (given in Orčić et al. [24]) and the sample dilution. DW, dry weight.” What does this mean exactly?

- Tables 2 and 3: No. You must give the exact values of all the compounds. There are other ways and methods to do this. You can only write here if the compound is not present. In this sense, I do not understand the method you used for this qualitative and quantitative analysis.

- “Of the quantified compounds, only quinic acid, amentoflavone and ferulic acid were present in all analyzed pulses samples.”. No. There are values everywhere.

 - Are all the values in Tables 2 and 3 normal or not for your seeds? Make comparisons reporting the values also with the same seed collected in other areas of the world. Most of all, you must specify if all these results are good or not from the nutraceutical point of view. A discussion on this last point is compulsory.

- Lines 197-201: Ok. Thus, you must detail these data for your seeds.

- Lines 221-222: Ok. Thus, you must detail these data for your seeds.

- Are all the values in Table 4 normal or not for your seeds? Make comparisons reporting the values also with the same seed collected in other areas of the world. Most of all, you must specify if all these results are good or not from the nutraceutical point of view. A discussion on this last point is compulsory.

- For all your tables, what about comparing your values with other species of the same genera?

- Lines 275-277: Values must be given here.

- The names of the species must be completely written the first time you cite them.

- Lines 294 on: You must explain all these correlations.

- Lines 343-344: In what sense and how?

MATERIALS AND METHODS:

- Homogenised? How? Why not fresh?

- When did you collect all these seeds and how much of them?

- For what concerns the purchased seeds, you must provide the geographical coordinates of their collection site as well as the altitude and the time of the collection.

 - For all the seeds, who performed the botanical identification of all the seed and how?

- For all the seeds, you must provide the voucher numbers for the collections.

- Line 393: A brief description of this method must be reported here.

- Where are all the data from the HPLC-MS/MS analysis i.e., retention time, mass values and mass fragments of your compounds?

- What about the standard compounds for this HPLC analysis? Where did you obtain them from?

- What about the gradient program?

- How did you prepare the extracts for the HPLC analysis?

- Lines 396-399: Really? Dilution with the eluting system?  

- v/v must be written in Italics.

CONCLUSIONS:

- Lines 442-443: What does this exactly mean?

- This section must be developed following my previous suggestions.

REFERENCES:

- Some references are incomplete.

Reviewer 2 Report

The manuscript studied the nutrients, bioactive compounds and elements of pulses. It was well organized and well written and of interest to the readers of Plants.

Reviewer 3 Report

Thank you -  the work is very readable. The work is factual and as such could you provide added sections that link what is there in a seed to nutritional value  eg bioavailability etc

Your introduction discusses problems with quality foods for the world -  thus a section in the conclusion etc as to the connection between the facts and this important factor would add value to the paper. 

Some little things that happen to be in my head are added as sticky notes-  may be topics to expand a little on. 

Round 2

Reviewer 1 Report

The authors presented a revised version of the paper Ihave previously reviewed.

My major concerns have been appropriately addressed now except one:

- I still think the data relative to the retention time, mass values and mass fragments of your compounds must be provided

Beside this, there are still two minor things:

- Line 14: Anti(nutrients). I meant erase the parentheses.

- Line 33: Pulses should not be in Italics.

All the rest is fine and after this, the manuscript can be accepted.

Round 3

Reviewer 1 Report

The authors presented a revised version of the paper Ihave previously reviewed.

My major concerns have been appropriately addressed now.

Thus, the manuscript can be accepted in its present form.